# Expanding the Chromosomal Evolution Understanding of Lygaeioid True Bugs (Lygaeoidea, Pentatomomorpha, Heteroptera) by Classical and Molecular Cytogenetic Analysis

**DOI:** 10.3390/genes14030725

**Published:** 2023-03-15

**Authors:** Natalia V. Golub, Anna Maryańska-Nadachowska, Boris A. Anokhin, Valentina G. Kuznetsova

**Affiliations:** 1Department of Karyosystematics, Zoological Institute, Russian Academy of Sciences, Universitetskaya emb. 1, 199034 St. Petersburg, Russia; 2Institute of Systematics and Evolution of Animals, Polish Academy of Sciences, Sławkowska 17, 31-016 Kraków, Poland

**Keywords:** karyotype, chromosome number, sex chromosomes, m-chromosomes, spermatocyte meiosis, FISH, 18S rDNA, (TTAGG)*_n_*, Lygaeoidea, Hemiptera

## Abstract

The Lygaeoidea comprise about 4660 species in 790 genera and 16 families. Using standard chromosome staining and FISH with 18S rDNA and telomeric (TTAGG)*_n_* probes, we studied male karyotypes and meiosis in 10 species of Lygaeoidea belonging to eight genera of the families Blissidae, Cymidae, Heterogastridae, Lygaeidae, and Rhyparochromidae. Chromosome numbers were shown to range from 12 to 28, with 2n = 14 being predominant. All species have an XY system and all but one have a pair of m-chromosomes. The exception is *Spilostethus saxatilis* (Lygaeidae: Lygaeinae); in another species of Lygaeinae, *Thunbergia floridulus*, m-chromosomes were present, which represents the first finding for this subfamily. All species have an inverted sequence of sex chromosome divisions (“post-reduction”). The 18S rDNA loci were observed on one or both sex chromosomes in *Kleidocerys resedae* and *Th. floridulus*, respectively (Lygaeidae), while on an autosomal bivalent in all other species. The rDNA loci tended to be close to the end of the chromosome. Using (TTAGG)*_n_*—FISH, we were able to show for the first time that the Lygaeoidea lack the canonical “insect” telomere motif (TTAGG)*_n_*. We speculate that this ancestral motif is absent from the entire infraorder Pentatomomorpha being replaced by some other telomere repeat motif sequences.

## 1. Introduction

The hemipteran superfamily Lygaeoidea (Heteroptera: Pentatomomorpha) is one of the largest and most diverse true bug groups comprising more than 4660 described species worldwide classified into about 790 genera and 16 families [1,2]. The superfamily is one of the cytogenetically better-studied groups of Pentatomomorpha, with basic cytogenetic data available for about 440 species [3,4,5,6,7,8,9,10,11,12]. The studied species represent about 9% of the total number of described lygaeoid species. Most of the species belong to the families Rhyparochromidae (175 species), Lygaeidae (128), and Blissidae (41), while for other families, data are sparse (Artheneidae, Berytidae, Colobathristidae, Cymidae, Heterogastridae, Geocoridae, Malcidae, Oxycarenidae, Pachygronthidae, and Piesmatidae) or absent altogether (Cryptorhamphidae, Meschiidae, and Ninidae).

Based on the current state of knowledge, lygaeoids exhibit a wide cytogenetic diversity in chromosome numbers and sex chromosome systems, sometimes even between closely related species. In general, chromosome numbers range from 10 to 46 (with some gaps) in diploid karyotypes; however, these extreme counts are not characteristic of the Lygaeoidea being found only sporadically, the lowest in Rhyparochromidae and Arhteneidae, while the highest in Berytidae [3,6,13,14]. In different lygaeoid families, both simple sex chromosome systems, XX/X(0) and XX/XY, and multiple systems, XnXn/XnY and XX/XYn, occur. In addition, a pair of m-chromosomes is encountered in all explored lygaeoid families, but are not found in the Berytidae and Piesmatidae families [8,14,15]. Despite the aforementioned diversity, the great majority of lygaeoids have 2n = 16, XY or 14, XY, and in many cases, the 16-chromosome karyotypes appear to be derivative of the original ones with 2n = 14, XY [4]. The course of male meiosis in lygaeoids conforms to the general heteropteran type as described in [3], being characterized, among other things, by the so-called sex chromosome “post-reduction”, with the sex chromosomes segregating equationally during the first round of meiosis, while separating reductionally during the second round of meiosis.

With rare exceptions, karyotype analysis was done using the routine chromosomal staining technique. Just a few lygaeoid species have been studied after DAPI/CMA_3_ fluorochrome staining and/or chromosome banding, including C-banding or NOR-banding [8,9,11,16,17,18]. Fluorescence in situ hybridization (FISH) has proven to be an effective and precise tool for physical mapping specific DNA sequences throughout the chromosomes [19]. There are also isolated examples of using FISH to detect and localize the major rDNA (28/26S, 18S, and 5.8S, spliced from a single precursor) on chromosomes of some lygaeoid species [11,17]. In all cases reported, rDNA loci were detected on one of the autosomal bivalents, but presumably not the same in different species. Finally, Grozeva et al. [17] undertook a dot-blotting experiment on *Oxycarenus lavaterae* (Fabricius, 1787) (Oxycarenidae) with different telomeric probes, the “insect” (TTAGG)*_n_* probe, and six alternative telomerase-based repeats; however, none of the tested probes hybridized with telomeres of the studied species.

In the present research, we obtained new data for ten species belonging to five families of Lygaeoidea. We studied for the first time the standard karyotypes of *Elasmolomus squalidus* (Gmelin, 1790) from the family Rhyparochromidae; *Nerthus* sp. 1 and *Nerthus* sp. 2 from the Heterogastridae; *Thunbergia floridulus* (Distant, 1918) and *Nysius cymoides* (Spinola, 1837) from the Lygaeidae. To identify new cytogenetic markers, we used FISH with the 18S rDNA probe to characterize the distribution profile of 18S rDNA in all but one (*Nerthus* sp. 1) above species and, additionally, in *Cymus claviculus* (Fallen, 1807) (Cymidae), *Dimorphopterus spinolae* (Signoret, 1857) (Blissidae), *Kleidocerys resedae* (Panzer, 1793), *Spilostethus saxatilis* (Scopoli, 1763), and *Nysius helveticus* (Herrich-Schäffer, 1850) (Lygaeidae). It is currently believed that Pentatomomoprha as a whole lack the canonical “insect” telomeric repeat TTAGG and, most likely, have a different mechanism of telomere elongation [20,21]; however, data for the superfamily Lygaeoidea, with the exception of the aforementioned dot-blotting experiment, are still missing. To fill this gap, we performed FISH mapping of the insect-type telomeric sequence (TTAGG)*_n_* on the chromosomes of all species explored.

## 2. Materials and Methods

The material used for karyotype analysis is listed in Table 1. Specimens (only males) were fixed immediately after collection in ethanol/acetic acid fixative (3:1) and stored in the fixative at 4 °C until slides were made. Species identification was made by Viktor B. Golub and Dmitry A. Gapon. Chromosome preparations were obtained from the testes of adult males using a squash technique as described elsewhere (e.g., [22]). The number of specimens karyotyped varied from one to three in the studied species, and several slides were prepared from the testes of each male. The standard karyotypes were studied after staining by the Schiff–Giemsa method [22,23]. To study the chromosomal distribution of the major rDNA, and to determine if the species display or not the “insect” telomere (TTAGG)*_n_* motif, double-target FISH with a firebug (*Pyrrhocoris apterus* (Linneus, 1758)) 18S rDNA probe and the (TTAGG)*_n_* probe was carried out according to the protocol described by Grozeva et al. [24]. As a positive control for the TTAGG telomeric probe, a barklouse (Psocomorpha) species, *Psococerastis gibbosa* (Sulzer, 1766), which is proven to be TTAGG-positive [25], was used. The whole procedure can be found in Golub et al. [23,25]. Standard preparations were photographed under oil immersion (X100 objective) using an Olympus BX 51 light microscope with an Olympus C-35 AD-4 camera (Olympus Optical Co., Tokyo, Japan). FISH preparations were photographed under oil immersion (X100 objective) using a Leica DM 6000 B microscope, Leica DFC 345 FX camera, and Leica Application Suite 3.7 software with an Image Overlay module (Leica Microsystems, Wetzlar, Germany). The filter sets applied were A, L5, and N21 (Leica Microsystems). The specimens, from which chromosome preparations were made, and the preparations themselves are stored at the Zoological Institute RAS (St. Petersburg, Russia).

## 3. Results

### 3.1. Family Rhyparochromidae

#### *Elasmolomus squalidus*, 2n = 12(8A + 2m + XY) (Figure 1a,b)

The karyotype was studied for the first time. At spermatocyte metaphase I (MI), there are five bivalents of autosomes, including a very small and negatively heteropycnotic pair of m-chromosomes, and two sex chromosomes, X and Y (meioformula: n = 4AA + mm + X + Y). One of the bivalents is noticeably larger than the others, which, with the exception of the m-chromosomes, make up a decreasing size range. The sex chromosomes are of different sizes—the larger one is taken as the X and the smaller one as the Y; they are located separately from each other, and each is split into chromatids (Figure 1a,b). The FISH using the 18S rDNA probe provided distinct signals on the second-largest bivalent (AA2); their intrachromosomal location remained unclear; no signals of the TTAGG telomeric probe were detected (Figure 1b).

### 3.2. Family Heterogastridae

#### 3.2.1. *Nerthus* sp. 1, 2n = 16(12A + 2m + XY) (Figure 1c)

The karyotype was studied for the first time; only Schiff-Giemsa staining was applied. At spermatocyte MI, there are 7 bivalents of autosomes, including a negatively heteropycnotic pair of m-chromosomes, and two sex chromosomes, X and Y (meioformula: n = 6AA + mm + X + Y). The autosomal bivalents, with the exception of the m-chromosomes, make up a decreasing size range. The sex chromosomes are of different sizes—they are located separately from each other, and each is split into chromatids (Figure 1c).

#### 3.2.2. *Nerthus* sp. 2, 2n = 18(14A + 2m + XY) (Figure 1d)

The karyotype was studied for the first time and only the FISH technique was applied. At spermatocyte MI, there are eight bivalents of autosomes, including a negatively heteropycnotic pair of m-chromosomes, and two sex chromosomes, X and Y (meioformula: n = 7AA + mm + X + Y). The autosomal bivalents, with the exception of the m-chromosomes, make up a decreasing size range. The sex chromosomes are of different size; they are located separately from each other and each is split into chromatids. FISH using the 18S probe provided distinct signals on the largest bivalent (AA1); their intrachromosomal location remained unclear; no signals of the TTAGG telomeric probe were detected (Figure 1d).

### 3.3. Family Cymidae

#### *Cymus claviculus*, 2n = 28(24A + 2m + XY) (Figure 1e–g)

The karyotype was previously studied by Pfaler-Collander from Finland [26], and our observations corroborate with those data. At spermatocyte MI, there are 13 bivalents of autosomes, including a negatively heteropycnotic pair of m-chromosomes, and two sex chromosomes, X and Y (meioformula: n = 12AA + mm + X + Y). The autosomal bivalents, with the exception of the m-chromosomes, make up a decreasing size range. The sex chromosomes are of different size sizes—they are split into chromatids and could be seen either next to each other (Figure 1e) or separately from one another (Figure 1f). FISH using the 18S probe provided distinct terminal signals on the AA1 bivalent at MI (Figure 1f) and on the two largest chromosomes in a spermatogonial cell (Figure 1g); no signals of the TTAGG telomeric probe were detected (Figure 1f,g).

### 3.4. Family Blissidae

#### *Dimorphopterus spinolae* 2n = 14(10A + 2m + XY) (Figure 1h–k)

The karyotype of *D. spinolae* was previously studied by Muramoto [27] who described the karyotype of males from Japan as “2n = 16 (and XY type)”, which most likely means 2n = 16(14 + XY). According to our observations, males collected in Voronezh region (Russia) have a different karyotype. At spermatocyte diakinesis/MI of these males, we observed six bivalents of autosomes, including a minute-sized m-chromosome pair, and two sex chromosomes, X and Y (meioformula: n = 5AA + mm + X + Y). One of the bivalents is noticeably larger than the others, which in turn decrease in size linearly. The sex chromosomes are of different sizes—they are split into chromatids and located separately from one another (Figure 1h,j). The sister metaphases II (MII) show six autosomes each, including the m-chromosome, and an XY-pseudobivalent; this latter and the m-chromosome are located in the center of the radial metaphase plate (Figure 1i). FISH using the 18S probe provided distinct terminal signals on the AA1 bivalent at MI (Figure 1j) and on the two largest chromosomes in spermatogonial cells (Figure 1k); no signals of the TTAGG telomeric probe were detected (Figure 1j,k).

### 3.5. Family Lygaeidae

#### 3.5.1. Subfamily Ischnorhynchinae

##### *Kleidocerys resedae*, 2n = 14(10A + 2m + XY) (Figure 2a–d)

The karyotype was previously studied by Pfaler-Collander from Finland [26], and our observations corroborate with those data. At spermatocyte MI, there are 6 bivalents of autosomes, including a negatively heteropycnotic pair of m-chromosomes, and two sex chromosomes, X and Y (meioformula: n = 5AA + mm + X + Y). The autosomal bivalents, with the exception of very small m-chromosomes, are poorly differentiated in size. The sex chromosomes are of different sizes—they are located separately from each other, and each is split into chromatids (Figure 2a,c). The sister metaphases II (MII) show each 6 autosomes, including the m-chromosome, and an XY-pseudobivalent; this latter and the m-chromosome are located in the center of the radial metaphase plate (Figure 2b). FISH using the 18S probe provided distinct terminal signals on each of the sex chromosomes at MI (Figure 2c) and in the spermatogonial nucleus (Figure 2d); no signals of the TTAGG telomeric probe were detected (Figure 2c,d).

#### 3.5.2. Subfamily Lygaeinae

##### *Spilostethus saxatilis* 2n = 14(12A + XY) (Figure 2e,f)

The karyotype was previously studied by Geitler [28], and our observations corroborate with those data. During the spermatocyte diakinesis/MI transition (Figure 2e,f), six bivalents of autosomes and two sex chromosomes, X and Y, were observed. No m-chromosomes were detected (meioformula: n = 6AA + X + Y). The bivalents make up a decreasing size range. The sex chromosomes are of different sizes—they are located separately from each other, and each is split into chromatids. FISH using the 18S probe provided distinct interstitial signals on each of the homologues of a medium-sized bivalent (AA); no signals of the TTAGG telomeric probe were detected (Figure 2f).

##### *Thunbergia floridulus* 2n = 16(12A + 2m + XY) (Figure 2g,h)

The karyotype was studied for the first time. At spermatocyte MI, there are seven bivalents of autosomes, including a negatively heteropycnotic pair of m-chromosomes, and two sex chromosomes, X and Y, each being split into the chromatids (meioformula: n = 6AA + mm + X + Y). The bivalents make up a decreasing size range; the sex chromosomes are of similar size and they are located separately from one another (Figure 2g,h). FISH, when using the 18S probe provided distinct signals on the putative X-chromosome; their intrachromosomal location remained unclear; no signals of the TTAGG telomeric probe were detected (Figure 2h).

#### 3.5.3. Subfamily Orsillinae

##### *Nysius cymoides*, 2n = 14(10A + 2m + XY) (Figure 2i,j)

The karyotype was studied for the first time. At spermatocyte MI, there are six bivalents of autosomes, including a negatively heteropycnotic pair of m-chromosomes, and two sex chromosomes, X and Y, each being split into the chromatids (meioformula: n = 5AA + mm + X + Y) (Figure 2i). One of the bivalents is very large, much larger than the second-largest bivalent. The sex chromosomes differ in size, and the putative Y chromosome is quite the same size as m-chromosomes. The latter were found to place separately in prophases (Figure 2j) while forming a bivalent at MI (Figure 2i). Sex chromosomes were located separately from each other at both MI and diakinesis (Figure 2i,j). Fluorescent signals of the 18S probe could be seen at terminal position on each homologue of the AA1 bivalent; no signals of the TTAGG telomeric probe were detected (Figure 2j).

##### *Nysius helveticus*, 2n = 14(10A + 2m + XY) (Figure 2k,l)

The karyotype was previously studied by Pfaler-Collander from Finland [26], and our observations corroborate with those data. Like in *N. cymoides*, at spermatocyte MI of this species, there are six bivalents of autosomes, including a negatively heteropycnotic pair of m-chromosomes, and two sex chromosomes, X and Y, each of which is split into the chromatids (meioformula: n = 5AA + mm + X + Y) (Figure 2k,l). One of the bivalents is very large, much larger than the second-largest bivalent. The sex chromosomes noticeably differ in size, and the smaller one, the putative Y chromosome, is quite the same size as the m-chromosomes. Sex chromosomes were located separately from each other at both diakinesis/MI transition and MI (Figure 2k,l). Fluorescent signals of the 18S probe could be seen at terminal position on each homologue of the AA1 bivalent; no signals of the TTAGG telomeric probe were detected (Figure 2l).

## 4. Discussion

### 4.1. Standard Karyotypes

In the present study, we have contributed to cytogenetics of the superfamily Lygaeoidea, the second-largest superfamily in the infraorder Pentatomomorpha, by examining ten different lygaeoid species from five families (Table 2). Our study confirms the previously published information that *Cymus claviculus* (Cymidae) has 28(24A + 2m + XY), *Kleidocerys resedae* (Lygaeidae: Ischnorhynchinae) − 14(10A + 2m + XY), *Spilostethus saxatilis* (Lygaeidae: Lygaeinae) − 14(12A + XY), and *Nysius helveticus* (Lygaeidae: Orsillinae) − 14(10A + 2m + XY). The high-numbered karyotype in *C. claviculus* was expected, since the family Cymidae is characterized by one of the highest chromosome numbers in the Lygaeoidea. In this family, chromosome numbers vary between 22 and 30, with the exception of the monotypic genus *Ninus* Stal, 1860 showing 2n = 16(12A + 2m + XY), which, along with 2n = 14(10A + 2m + XY), is the most commonly found in the family as a whole [3,4,6]. *K. resedae* and *S. saxatilis* have the same karyotypes as all other so far studied species within their genera [3,4]. *Nysius helveticus* has a karyotype found in the vast majority of species of the genus [3,5]. On the other hand, Muramoto [27] recorded *Dimorphopterus spinolae* in Japan as having “2n = 16 (and XY type)”, which we interpret as 16(14A + XY); however, this does not match our observations of this species karyotype. In all studied males of *D. spinolae* collected from Voronezh region of Russia, we found 2n = 14(10A + 2m + XY). Since Muramoto did not provide a photograph or a drawing of the karyotype, it is difficult to explain the discrepancy between the data.

The karyotypes of five species were studied herein for the first time. *Elasmolomus squalidus* (Rhyparochromidae) was found to share 2n = 12(8A + 2m + XY) with *E. sordidus* (Fabricius, 1787) [29] and *Elasmolomus* sp. [30], while two other species studied in this genus, *E. mendicus* Stål, 1872 and *E. transversus* (Signoret, 1860), have 2n = 14(10A + 2m + XY) [4]. In the subfamily Orsillinae (Lygaeidae), *Nysius cymoides* have 2n = 14(10A + 2m + XY), just like the aforementioned *N. helveticus*. The same karyotype was previously recorded for all studied species of this genus, with only two exceptions: *N. senecionis* (Schilling, 1829) and *N. tennelus* Barber, 1947 were reported to have a karyotype 2n = 22(18A + 2m + XY) diverging much from the modal one in terms of the number of autosomes [3,5].

Our data on *Thunbergia floridulus* (Lygaeidae: Lygaeinae) and *Nerthus* spp. (Heterogastridae) are the first for their genera. *Th. floridulus* was found to have 2n = 16(12A + 2m + XY), and it is the first species with m-chromosomes discovered in the subfamily Lygaeinae. Our data discredit the early idea that this subfamily does not have, unlike all other subfamilies of the Lygaeidae, m-chromosomes [3,5,13]. Moreover, given the presence of m-chromosomes in all families of the Lygaeoidea, we can now suggest that m-chromosomes were already present in the ancestral lygaeoid karyotype and disappeared independently in individual species and higher taxa of this superfamily.

The data obtained herein on the small genus *Nerthus* Distant, 1909 turned out to be intriguing. The two males collected from an undetermined bush in the Popa mount, Myanmar, and fixed for chromosome analysis were identified as *N. dudgeioni* Distant, 1909. However, these males showed different karyotypes, 16(14A + 2m + XY) and 18(14A + 2m + XY), thus casting doubt on whether they belong to the same species. At this stage, we therefore designated them as *Nerthus* sp. 1 and *Nerthus* sp. 2. The solution to this problem remains a challenge for future research.

Our data cover five different families of the Lygaeoidea and all three recent subfamilies of the largest family Lygaeidae. Despite the small number of studied species, we discovered five different chromosome counts: 12 (1 species), 14 (5), 16 (2), 18 (1), and 28 (1), and six different karyotypes: 12(8A + 2m + XY), 14(12A + XY), 14(10A + 2m + XY), 16(12A + 2m + XY), 18(14A + 2m + XY), and 28(24A + 2m + XY). This indicates that the Lygaeoidea have a wide diversity in chromosome numbers, which is consistent with previous analyses that, in total, used far more species than the current research (reviewed in [6]). As discerned in previous reviews, the karyotypes 2n = 14(10A + 2m + XY) and 2 = 16(12A + 2m + XY), which predominate in our study, clearly predominate in the Lygaeoidea as a whole [3,5,6,13,16]. In many cases, the 16-chromosome karyotypes appear to be derivatives of the putative ancestral karyotype with 2n = 14 [5,13].

#### 4.1.1. Spermatocyte Meiosis

As stated in the Introduction, male meiosis in the Heteroptera is unique, showing an inverted sequence of sex chromosome divisions (“post-reduction”), with sex chromosomes undergoing equational separation of sister chromatids during the first division and reductional segregation of sex chromosomes during the second division; the autosomes remain their conventional sequence of divisions [3]. Although exceptions are known and some true bug taxa have sex chromosome “pre-reduction” (see for references [3]; see also, e.g., [12,22,23,31,32]), all the studied species of the Lygaeoidea demonstrate the “post-reduction”. This is also the case with the species studied in the present work, which is most convincingly shown in *Dimorphopterus spinolae* and *Kleidocerys resedae*, in which it was possible to trace the behavior of sex chromosomes up to MII. In these species, X- and Y-chromosomes were located separately from each other and split into chromatids at MI (as with all other species). At MII, the X and Y were observed to come together and form a pseudobivalent in the spindle. Unfortunately, in our material, there was not a second anaphase (AII) in which the sex chromosomes segregated, thus demonstrating the whole process called the “touch-and-go pairing” [3].

#### 4.1.2. 45S rDNA-FISH

The 45S major rDNA cistron is known to be highly dynamic in size, number, and distribution of chromosomal loci in the genome. This can be accomplished through different mechanisms, including an increase in the number of copies, chromosomal rearrangements, such as translocations, fusions, and inversions, ectopic recombination between non-homologous chromosomes, unequal exchanges, and transposition of rDNA repeats to new locations followed by amplification or deletion of original loci [33,34]. The number and chromosomal distribution of rDNA loci in the karyotype detected by FISH provide additional chromosomal landmarks useful for understanding the genome structure and evolution in different insect taxa. True bugs have holokinetic chromosomes in which centromeres are absent, so the search for chromosomal markers is of particular importance in this group. In several species, FISH-based karyotyping, specifically rDNA mapping, has pointed to chromosomal differences between species with similar karyotypes [22,35]. In recent years, studies using FISH, mainly for analyzing the 18S region of the major 45S rDNA, have become very popular in true bugs (reviewed in [24,25,36,37,38,39] and references therein). However, such studies have rarely been attempted in the Lygaeoidea. So far, physical mapping of 18S has only been done in *Oxycarenus lavaterae* (Fabricius, 1787) from the family Oxycarenidae [17] and in three species from the family Lygaeidae, *Ochrimnus sagax* Brailovsky, 1982, *Oncopeltus femoralis* Stal, 1874, and *Lygaeus peruvianus* Brailovsky, 1978, all from the subfamily Lygaeinae [11]. In all these species, 18S loci were detected on autosomal bivalents, and their chromosomal location was not uniform. For example, in *O. sagax*, it was the largest bivalent, while in all other species, the 18S-bearing bivalent has been identified as “an autosomal pair”, meaning it was most likely not the largest one. It is also stated that *O. femoralis* and *L. peruvianus* have an interstitial chromosomal position of 18S repeats.

In the present study, we analyzed 18S chromosomal positions in nine more species (*Nerthus* sp. 1 has remained unstudied), eight more genera, four more families of the Lygaeoidea, and, in addition, we obtained new data for each of the three subfamilies of the family Lygaeidae (Table 2). Our data demonstrated that the spatial positioning of ribosomal genes in the lygaeoid true bugs is sufficiently diverse. Autosomal location was found to predominate being found in seven of the nine studied species and in each of the families, but the autosomes bearing *rRNA* genes turned out to be different in different species. Specifically, the rDNA loci are located on a medium-sized bivalent in *Spilostethus saxatilis*, on the second-largest bivalent in *Elasmolomus squalidus* and on the largest bivalent in all other species (regardless of the number of chromosomes they possess). Moreover, we have identified two deviant patterns of the rDNA markings, both in the family Lygaeidae, namely, on both X- and Y- chromosomes in *K. resedae* (Ischnorhynchinae) and on one of the sex chromosomes, supposedly the X, in *Th. floridulus* (Lygaeinae).

In addition, the intrachromosomal location of *rRNA* genes does not seem to be the same in different species, although the location close to the chromosome ends predominates. In those species in which we had the opportunity to observe mitotic chromosomes, such as *Cymus claviculus*, *Dimorphopterus spinolae*, and *Kleidocerys resedae*, the ribosomal gene clusters were visible at the terminal position. In the other six species, the analysis was done mainly at MI when the chromosomes are highly condensed, preventing the accurate determination of the position of hybridization signals. However, at least in *Spilostethus saxatilis*, they seem to be located in an interstitial region in the chromosomes.

The autosomal distribution of *rRNA* genes is found in different non-related families, such as Heterogastridae, Cymidae, Blissidae, and Lygaeidae, as well as previously studied Oxycarenidae. Despite the limited sample of studied species, we can hypothesize that the autosomal location is an ancestral trait in the superfamily Lygaeoidea. Moreover, the autosomal rDNA location pattern predominates over all other patterns in other families of Pentatomomorpha (see, e.g., [37,40]); therefore, we can assume its ancestry in the infraorder as a whole. The divergent patterns found in our study in the subfamilies Ischnorhynchinae (*K. resedae*) and Lygaeinae (*Th. floridulus*) can be regarded as having arisen de novo and independently in the evolution of the family Lygaeidae. Future studies involving more lygaeoid taxa will allow for testing this hypothesis.

In the current literature, it is widely debated whether ribosomal genes are distributed randomly within chromosomes or, on the contrary, their distribution is regulated by some special mechanisms, and the rDNA loci tend, therefore, to occur preferentially in one or another region on the chromosome [41,42]. Recent analyses show that, although rDNA may actually occur in different chromosomal positions, there are some trends in particular groups of animals. For example, within three very large and most extensively studied “monocentric” insect orders, Coleoptera, Hymenoptera, and Orthoptera, the terminal (in the first) and pericentromeric or interstitial (in the last two) locations of the rDNA clusters seem to be more frequent [40,43,44,45]. In insects with holokinetic chromosomes, only two alternative intrachromosomal positions of rDNA, terminal and interstitial, can be expected and both have been detected [46,47,48,49], with terminal position (always on X chromosomes) being a general feature of aphids ([50] and references therein). In the superfamily Lygaeoidea, both patterns were observed or, at least, suspected, although in all more conclusively confirmed cases, it is a terminal location. Some reviews and some original studies (e.g., [24,36,39]) suggest the terminal pattern to be more abundant in true bugs; however, there has never been any targeted study of this problem and, therefore, the question of whether this pattern is a real trend in these insects remains open. As already noted above, all rDNA-FISH analyses in true bugs have almost exclusively been carried out in meiosis, mainly at MI, and the compressed structure and small size of meiotic bivalents could prevent the accurate determination of loci positions in the majority of cases [22].

#### 4.1.3. (TTAGG)*_n_*-FISH

In none of the species studied herein, the (TTAGG)*_n_* probe labeled any chromosome areas, indicating the absence of the 5 bp repeat (TTAGG)*_n_*, the commonest and supposedly an ancestral DNA motif of insect telomeres [20,21]. Although data are only available for nine species, they cover five differently related families of the superfamily Lygaeoidea. No data are yet available for 11 minor families. Earlier, based on data available at that time for only a few species of the family Pentatomidae [17,51,52,53], it was speculated that the insect motif (TTAGG)*_n_* is absent (lost) in the entire clade Pentatomomorpha [17,20,53]. Our data reinforce thus this hypothesis. Until recently, the question remained unanswered as to whether this motif had been replaced with any other motif or whether Pentatomomorpha had acquired some an alternative telomerase-independent mechanism of telomere maintenance. Recently, the first chromosome-level telomere-to-telomere genome assemblies have been achieved for *Acanthosoma haemorrhoidale* (Linnaeus, 1758) from the family Acanthosomatidae and *Aelia acuminata* (Linnaeus, 1758) from the family Pentatomidae, both from the infraorder Pentatomomorpha, and the distribution of telomere repeat sequences in the assembled genomes of these species was analyzed [54]. The results showed that *A. haemorrhoidale* and *A. acuminata* contained at the telomeres the 10 bp repeats, TTAGGGATGG and TTAGGGTGGT, respectively. The DNA-based approach proved to be an efficient tool for identifying yet unknown telomere motifs. Specifically with regard to the Pentatomomorpha true bugs, it is now possible to assume that they have, instead of the consensus “insect” telomere motif (TTAGG)*_n_*, some other repeat motifs at their chromosome termini.

## 5. Conclusions

A study of 10 species of Lygaeoidea (Pentatomomorpha) showed that they have a considerable variety of chromosome numbers ranging from 12 to 28 (specifically, 12–18, and 28) in male diploid karyotypes, with 2n = 14 being predominant. A pair of m-chromosomes is present in all but one species, and m-chromosomes are, for the first time, found in the subfamily Lygaeinae (Lygaeidae). All species have an XY sex chromosome system and an inverted sequence of sex chromosome divisions (“post-reduction”) in male meiosis. The major rDNA can be located on both autosomes and sex chromosomes while the former pattern is predominant. The species studied do not have the “insect” telomere motif (TTAGG)*_n_*, which reinforces the hypothesis that this motif is not present in the entire infraorder Pentatomomorpha, being replaced in this group with some other telomere repeats.

## Figures and Tables

**Figure 1 genes-14-00725-f001:**
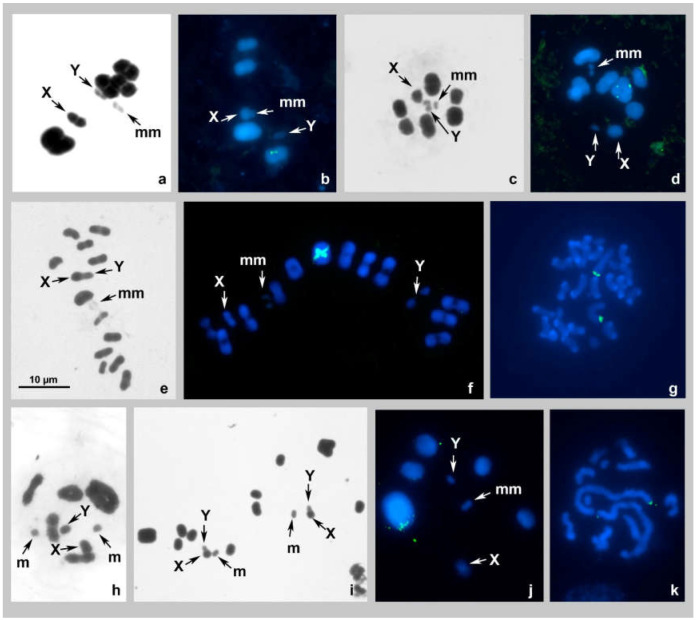
(**a**–**k**). Karyotypes of species of the families Rhyparochromidae (**a**,**b**), Heterogastridae (**c**,**d**), Cymidae (**e**–**g**), and Blissidae (**h**–**k**) after Schiff–Giemsa and double-target FISH with 18S rDNA and telomeric (TTAGG)_n_ probes. (**a**,**b**) *Elasmolomus squalidus* MI standard staining (**a**), MI FISH (**b**); (**c**) *Nerthus* sp. 1 MI standard staining; (**d**) *Nerthus* sp. 2 MI FISH; (**e**–**g**) *Cymus claviculus* MI standard staining (**e**), MI FISH (**f**) spermatogonial metaphase FISH (**g**); (**h**–**k**) *Dimorphopterus spinolae* late diakinesis standard staining (**h**), MII standard staining (**i**), MI FISH (**j**), and spermatogonial metaphase FISH (**k**). 18S rDNA signals are green; TTAGG signals are absent. Bar = 10 µm in all figures.

**Figure 2 genes-14-00725-f002:**
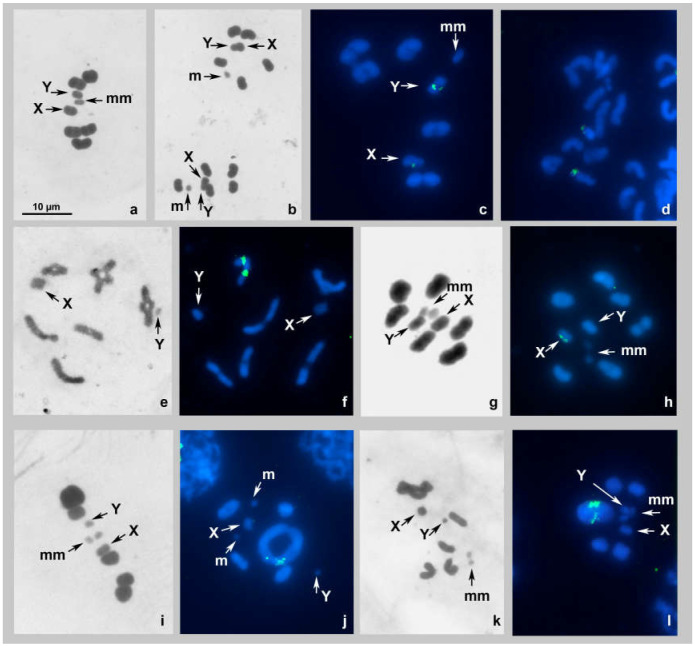
(**a**–**l**). Karyotypes of species of the family Lygaeidae after Schiff–Giemsa and double-target FISH with 18S rDNA and telomeric (TTAGG)*_n_* probes. (**a**–**d**) *Kleidocerys resedae* MI standard staining (**a**), MII standard staining (**b**), MI FISH (**c**), spermatogonial metaphase FISH (**d**); (**e**,**f**) *Spilostethus saxatilis* diakinesis/MI transition standard staining (**e**), diakinesis/MI transition FISH (**f**); (**g**,**h**) *Thunbergia floridulus* MI standard staining (**g**) MI FISH (**h**); (**i**,**j**) *Nysius cymoides* MI standard staining (**i**), diakinesis FISH (**j**); (**k**,**l**) *Nysius helveticus* diakinesis/MI transition standard staining (**k**), MI FISH (**l**). 18S rDNA signals are green; TTAGG signals are absent. Bar = 10 µm in all figures.

**Table 1 genes-14-00725-t001:** Localities of the material used for chromosome analysis.

Species	Number of Males Studied	Date and Place of Collection
	**Rhyparochromidae**	
*Elasmolomus squalidus* (Gmelin, 1790)	3	2–4 November 2019, Popa mount, Myanmar (D. Gapon leg.)
	**Heterogastridae**	
*Nerthus* sp. 1	1	2 November 2019, Popa mount, Myanmar (D. Gapon leg.)
*Nerthus* sp. 2	1	2 November 2019, Popa mount, Myanmar (D. Gapon leg.)
	**Cymidae**	
*Cymus claviculus* (Fallen, 1807)	2	16 August 2022, Voronezh region, Russia (V. Golub leg.)
	**Blissidae**	
*Dimorphopterus spinolae* (Signoret, 1857)	3	14 August 2022, Voronezh region, Russia (V. Golub leg.)
	**Lygaeidae**	
	**Ischnorhynchinae**	
*Kleidocerys resedae* (Panzer, 1793)	3	24 August 2022, Voronezh region, Russia (V. Golub leg.)
	**Lygaeinae**	
*Spilostethus saxatilis* (Scopoli, 1763)	3	25 June 2021, Sevan vic., Armenia (D. Gapon leg.)
*Thunbergia floridulus* (Distant, 1918)	2	13 November 2019, Pai, Thailand (D. Gapon leg.)
	**Orsillinae**	
*Nysius cymoides* (Spinola, 1837)	2	1 August 2022, Goryachiy Kluch vic., Russia (V. Golub leg)
*Nysius helveticus* (Herrich-Schäffer, 1850)	3	14–16 August 2022, Voronezh region, Russia (V. Golub leg.)

**Table 2 genes-14-00725-t002:** Karyotypes of the studied species, with data on chromosome number and diploid karyotype formulae, rDNA loci location, TTAGG repeats, and references. A—autosome, AA—autosomal bivalent.

Species	2n	18S rDNA Location	TTAGG Repeat	Published Data
**Rhyparochromidae**				
*Elasmolomus squalidus*	12(8A + 2m + XY)	AA2	Not found	No
**Heterogastridae**				
*Nerthus* sp. 1	16(12A + 2m + XY)	Not studied	Not studied	No
*Nerthus* sp. 2	18 (14A + 2m + XY)	AA1	Not found	No
**Cymidae**				
*Cymus claviculus*	28(24A + 2m + XY)	AA1 (close to the ends)	Not found	[26]: n = 12 + m + X + Y
**Blissidae**				
*Dimorphopterus spinolae*	14(10A + 2m+ XY)	AA1 (close to the ends)	Not found	[27]: “2n = 16 (and XY type)”
**Lygaeidae**				
**Ischnorhynchinae**				
*Kleidocerys resedae*	14(10A + 2m + XY)	X and Y (close to the end)	Not found	[26] (as *Ischnorhynchus lineatus*): 2n = 14, n = 5 + m + X + Y
**Lygaeinae**				
*Spilostethus saxatilis*	14(12A + XY)	AA (interstitially)	Not found	[28]: n = 6 + X(Y)
*Thunbergia floridulus*	16(12A + 2m + XY)	X	Not found	No
**Orsillinae**				
*Nysius cymoides*	14(10A + 2m + XY)	AA1 (close to the ends)	Not found	No
*Nysius helveticus*	14(10A + 2m + XY)	AA1 (close to the ends)	Not found	[26] (as *N. lineatus*): 2n = 14, n = 5 + m + X + Y

## Data Availability

The data presented in this study are available upon request from the corresponding author.

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
