# Peer review of "Expanding the Chromosomal Evolution Understanding of Lygaeioid True Bugs (Lygaeoidea, Pentatomomorpha, Heteroptera) by Classical and Molecular Cytogenetic Analysis"

_genes, 2023, doi:10.3390/genes14030725_

Round 1
Reviewer 1 Report
In the present study Golub and colleagues used classical and usual molecular cytogenetic probes (18S rDNA and TTAGG) to advance on the characterization of karyotypes from Lygaeoidea bugs. This is a quite well written manuscript with clear information, adding important data on the understanding of chromosomal evolution on these insects, allowing to fill gaps for specific groups. I have only a few criticisms on specific parts of the text that should be considered for better reading and to broaden interest in the work.
- Title: The title does not reflect the content of the manuscript and should be replaced for two reasons: (1) “Comparative chromosome analysis …”: In my opinion due to the analysis of distant species it was not possible to perform a clear comparative analysis of the data, but a general analysis on the superfamily, that is relevant and interesting to have broad ideas about chromosome evolution. “molecular characterization of …”: It was not a molecular characterization, but a chromosomal analysis. My suggestion of a tentative title is: “Expanding the chromosomal evolution understanding on lygaeioid true bugs (Lygaeoidea, Tentatomomorpha, Heteroptera) by classical and molecular cytogenetic analysis.”;
- Line 80: change “dot-blotting experiment” to “O. larvatae”;
- Results: the results section is long and with unnecessary repetition of sentences. My suggestion is combining all data in the description and the details should appear only on table 2. This is not mandatory, but with this change the reading of the results will be much easier;
- Results: please standardize the description of the karyotypes. In some parts it is described as the diploid karyotype and in others haploid. Also, in some parts it is AA and others A (see table 2);
- Line 126: replace “mm-bivalent” to “m-chromosomes”. Also, along the whole text;
- Lines 143, 183, 198, and 231: change “her/his” to “the”;
- Lines 212, 213: The occurrence of rDNA signals only on the Y chromosome is intriguing. How are the rRNA synthesized in females? At least some spread copies (not detected due to FISH resolution limit) should be present. Or the signals are in fact on the X chromosome, which due to the similar size with the Y chromosome could make the precise identification difficult. Please check on other cells to confirm this information;
- Line 221: delete “.”;
- Line 251: change “insights” to “information”;
- Line 292, 293: please delete the sentence “The solution of this problem remains a challenge for future research.”;
- Table 2: Please include a caption for this table;
- Lines 351, 352: “Our data demonstrated that the spatial positioning of ribosomal genes in the lygaeoid true bugs is sufficiently diverse.” To a broader interest of the discussion, I suggest the inclusion of information about rDNA diversity on other insects. There are some references that could be acknowledge on this part, see below:
1. (HEMIPTERA: CIMICOMORPHA): Pita S, Lorite P, Cuadrado A, Panzera Y, De Oliveira J, Alevi KCC, Rosa JA, Freitas SPC, Gómez-Palacio A, Solari A, Monroy C, Dorn PL, Cabrera-Bravo M, Panzera F. High chromosomal mobility of rDNA clusters in holocentric chromosomes of Triatominae, vectors of Chagas disease (Hemiptera-Reduviidae). Med Vet Entomol. 2022 Mar;36(1):66-80. doi: 10.1111/mve.12552. Epub 2021 Nov 3. PMID: 34730244.
2. (COLEOPTERA): Cabral-de-Mello DC, Oliveira SG, de Moura RC, Martins C. Chromosomal organization of the 18S and 5S rRNAs and histone H3 genes in Scarabaeinae coleopterans: insights into the evolutionary dynamics of multigene families and heterochromatin. BMC Genet. 2011 Oct 15;12:88. doi: 10.1186/1471-2156-12-88. PMID: 21999519; PMCID: PMC3209441.
3. (LEPIDOPTERA): Provazníková I, Hejníčková M, Visser S, Dalíková M, Carabajal Paladino LZ, Zrzavá M, Voleníková A, Marec F, Nguyen P. Large-scale comparative analysis of cytogenetic markers across Lepidoptera. Sci Rep. 2021 Jun 9;11(1):12214. doi: 10.1038/s41598-021-91665-7. PMID: 34108567; PMCID: PMC8190105.
4. (ORTHOPTERA): Cabrero J, Camacho JP. Location and expression of ribosomal RNA genes in grasshoppers: abundance of silent and cryptic loci. Chromosome Res. 2008;16(4):595-607. doi: 10.1007/s10577-008-1214-x. Epub 2008 Apr 26. PMID: 18431681.
5. (HEMIPTERA: CICADOMORPHA): Anjos A, Paladini A, Evangelista O, Cabral-de-Mello DC. Insights into chromosomal evolution of Cicadomorpha using fluorochrome staining and mapping 18S rRNA and H3 histone genes.Journal of Zoological Systematics and Evolutionary Research. 2018;57:314–322. https://doi.org/10.1111/jzs.12254
(HYMENOPTERA): Teixeira GA, de Aguiar HJAC, Petitclerc F, Orivel J, Lopes DM, Barros LAC. Evolutionary insights into the genomic organization of major ribosomal DNA in ant chromosomes. Insect Mol Biol. 2021 Jun;30(3):340-354. doi: 10.1111/imb.12699. Epub 2021 Mar 8. PMID: 33586259.
- Line: 393: change “open today” to “opened”;
- Lines 408-414: the reference is missing;
- Line 423: “m-chromosomes are FOR THE first time …”;
Author Response
Dear Reviewer,
Thank you for your time and valuable comments. All changes have been made according to your remarks. In the file attached, we provide a response to each of your comments. Please, use the file genes-2288678-coverletter_1
Kind regards,
Valentina Kuznetsova
On behalf of all the co-authors

Reviewer 2 Report
The manuscript deals with the study of insect telomeric repeat motif, location of 18SrDNA in the chromosomes and male meiosis in ten species belonging to five families of Lygaeoidea in the infraorder Pentatomomorpha in Heteroptera. In addition to findings in individual species the authors were able to provide additional evidence for the absence of the canonical insect telomeric repeat motif (TTAGG)n from Lygaeoidea and actually from the whole of Pentatomomorpha. This is, by far, the most significant observation of the study. The manuscript is mainly well written and concise. The quality of photographic documentation is excellent.
Remarks
1. line 11 …4,660 in 790 genera … preferably 4660 species in 790 genera..
2 the number 4660 without a hyphen throughout the manuscript
3 Line 76: Nyzius helveticus is mentioned here and in Table 1, but the name Nysius helveticus, is present in the results. Please correct throughout the manuscript. Actually, the authors should consider carefully when to use the entire name of the genus and when to abbreviate it (e.g., Nysius helveticus -> N. helveticus).
4 In results, the descriptions of chromosome numbers like (meioformula n=5AA+mm+X+Y) or 2n=14(10A+2m+XY) are clear and unambiguous, but are there any specific reasons for the description like 2n=12(8AA+2m+XY), if not please correct.
5. line 171-178, Figure text, species names are written with normal font, should be changed into italics
Author Response
Dear Reviewer,
Thank you for your time and for your valuable comments. All changes have been made according to your remarks. Please, find attached a point-by-point response to your comments.
Sincerely,
Valentina Kuznetsova
On behalf of all the co-authors
